# Predictability of Microbial Adhesion to Dental Materials by Roughness Parameters

**Andrea Schubert** \*,†, **Torsten Wassmann** \*,† 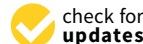, **Mareike Holtappels, Oliver Kurbad, Sebastian Krohn and Ralf Bürgers**

Department of Prosthodontics, University Medical Center Goettingen, Robert-Koch-Str. 40, 37075 Goettingen, Germany

\* Correspondence: andrea.schubert@med.uni-goettingen.de (A.S.); torsten.wassmann@med.uni-goettingen.de (T.W.)

† These authors contributed equally to this work.

**Abstract:** Microbial adhesion to intraoral biomaterials is associated with surface roughness. For the prevention of oral pathologies, smooth surfaces with little biofilm formation are required. Ideally, appropriate roughness parameters make microbial adhesion predictable. Although a multitude of parameters are available, surface roughness is commonly described by the arithmetical mean roughness value ($R_a$). The present study investigates whether $R_a$ is the most appropriate roughness parameter in terms of prediction for microbial adhesion to dental biomaterials. After four surface roughness modifications using standardized polishing protocols, zirconia, polymethylmethacrylate, polyetheretherketone, and titanium alloy specimens were characterized by $R_a$ as well as 17 other parameters using confocal microscopy. Specimens of the tested materials were colonized by *C. albicans* or *S. sanguinis* for 2 h; the adhesion was measured via luminescence assays and correlated with the roughness parameters. The adhesion of *C. albicans* showed a tendency to increase with increasing the surface roughness—the adhesion of *S. sanguinis* showed no such tendency. Although $S_a$, that is, the arithmetical mean deviation of surface roughness, and $R_{dc}$, that is, the profile section height between two material ratios, showed higher correlations with the microbial adhesion than $R_a$, these differences were not significant. Within the limitations of this in-vitro study, we conclude that $R_a$ is a sufficient roughness parameter in terms of prediction for initial microbial adhesion to dental biomaterials with polished surfaces.

**Keywords:** dental materials; roughness; roughness parameters; microbial adhesion; *Candida albicans*; *Streptococcus sanguinis*

## 1. Introduction

In the human oral cavity, teeth and artificial hard surfaces like implant abutments and denture bases are immediately covered by salivary components, and colonized by a multitude of microorganisms that may form biofilms. Oral biofilms are responsible for dental pathologies such as caries, periodontitis, peri-implantitis, denture stomatitis, or candidiasis [1–4]. The first and essential step in biofilm formation is the initial attachment of single microbes to a substratum [5]. In the initial process of adhesion, facultative anaerobic pioneer bacteria such as *Streptococcus sanguinis* (*S. sanguinis*) are pivotal for the subsequent adhesion of cariogenic and periodontal pathogens [6,7]. Besides bacteria, the facultative pathogenic yeast *Candida albicans* (*C. albicans*) is found on implant and denture base materials [8–12], which is associated with stomatitis [10,13], a local mucosal inflammation with a high incidence among denture wearers [14]. To prevent oral diseases, it is apparent that dental materials with a low susceptibility to biofilm formation are required. The microbial adhesion correlates to its surface

roughness (topography) and its surface free energy (SFE; hydrophobicity) after manufacturing [3]. Other surface properties, such as surface charge, substratum chemistry, and substratum stiffness, have a minor influence on microbial adhesion and subsequent biofilm formation [15,16]. Attempts to clinically minimize microbial adhesion to dental materials demand smooth surfaces with a low SFE [17]. In the literature, many parameters describe surface roughness, some of which are suitable for discriminating between the polished surfaces of dental materials [18–20]. The rapidly increasing number of parameters was called "the parameter rash" by Whitehouse, in 1982 already [21]. The arithmetical mean roughness value ($R_a$) is the most commonly used parameter for in vitro and in vivo surface roughness evaluation [22–27]. While $R_a$ describes the roughness along a two-dimensional profile, $S_a$ is the arithmetical mean deviation of surface roughness, and characterizes the surface roughness three-dimensionally [20]. Ideally, a surface parameter will predict microbial adhesion, and indicates the extent of the required surface processing before applying a material into the oral cavity. $R_a$ may not be the most appropriate parameter in this regard.

The aim of the current in-vitro study was to compare $R_a$ and 17 other two- or three-dimensional roughness parameters, for their prediction of adhesion of *C. albicans* and *S. sanguinis* to implant and abutment materials, as well as fixed and removable prosthesis materials. Surfaces of standardized zirconia (YTZP), polymethylmethacrylate (PMMA), polyetheretherketone (PEEK), and titanium alloy (Ti) specimens were characterized using confocal microscopy after four surface roughness treatments. The SFE was determined using the sessile-drop method for contact angles. The specimens were colonized with either *C. albicans* or *S. sanguinis*, and their adhesion was measured via a luminescence assay. Analyses of the adhesion were correlated with surface roughness parameters. We hypothesized that the three-dimensional surface roughness parameters would have a better correlation with the initial *C. albicans* and *S. sanguinis* adhesion to surfaces than the two-dimensional profile parameter $R_a$. The results of the present study will outline which surface roughness parameters are the most appropriate in order to predict the tendency for microbial adhesion to dental materials.

## 2. Materials and Methods

### 2.1. Specimen Preparation

Cylindrical specimens with a diameter of 10 mm and a height of 2.5 mm were manufactured (*n* = 21) from four representative dental materials (Table 1). Using computer-aided design (CAD)/computer-aided manufacturing (CAM) technology, according to manufacturer's instructions (Zirkonzahn, Gais, Italy), rods of YTZP, PEEK, and PMMA were produced; the rods of Ti were obtained prefabricated. The rods were sliced into disks using a separating machine (Micracut 201, Metkon, Bursa, Turkey).

**Table 1.** Specification of the materials used in this study.

| Product | Type | Manufacturer |
|---|---|---|
| Ice Zirkon 95H14 | yttria-stabilized zirconia ceramic (YTZP) | Zirkonzahn, Gais, Italy |
| Tecno Med 95H16 | polyetheretherketone (PEEK) | Zirkonzahn |
| PalaXPress | polymethylmathacrylate (PMMA) | Heraeus Kulzer, Hanau, Germany |
| Ti-6Al-4V | titanium alloy (Ti) | Hempel metals and more, Duebendorf-Zurich, Switzerland |

### 2.2. Sample Treatment and Roughness Measurements

Surface treatments of the specimens were performed with an automated grinding machine (Digiprep 251, Metkon) and silicon carbide grinding paper, with successively decreasing grain sizes (400, 800, 1200 and 4000). The specimen surfaces were modified to four topographic roughnesses represented by specific $R_a$ values (Table 2). The $R_a$ values were calculated for five specimens at three sites of each tested material via confocal microscopy (Zeiss Smartproof 5, Carl Zeiss, Jena, Germany)

and automated software analysis (ConfoMap ST, version 7.4.8076, Carl Zeiss). Likewise, 17 roughness parameters were calculated for each roughness level (Table 3). These parameters were chosen based on ISO 4287 [28] for the two-dimensional parameters, and ISO 25178 [29] for the three-dimensional parameters. The same specimens were tested successively—after completing the adhesion testing for one roughness level, the specimens were rinsed with distilled water and 70% ethanol, dried, and polished to the next lower $R_a$ value level. The microbial adhesion was tested as described below.

**Table 2.** Roughness levels with desired $R_a$ values of the tested specimens after surface treatment.

| Roughness Level | $R_a$ (µm) |
|:---:|:---:|
| I | <0.1 |
| II | ~0.2 |
| III | 0.7–1 |
| IV | 1.7–2 |

**Table 3.** Summary of the roughness parameters used in this study.

| Parameter | Symbol | |
|---|:---:|:---:|
| 1. Maximum peak height of the roughness profile | $R_p$ | |
| 2. Maximum valley depth of the roughness profile | $R_v$ | |
| 3. Mean roughness depth | $R_z$ | |
| 4. Mean height of profile elements | $R_c$ | |
| 5. Total height of the roughness profile | $R_t = R_{\max}$ | two-dimensional |
| 6. Arithmetical mean roughness value | $R_a$ | |
| 7. Root-mean-square roughness | $R_q = R_{\mathrm{ms}}$ | |
| 8. Skewness of the roughness profile | $R_{sk}$ | |
| 9. Kurtosis of the roughness profile | $R_{ku}$ | |
| 10. Profile section height between two material ratios | $R_{dc}$ | |
| 11. Material component of the profile | $R_{mr}$ | |
| 12. Root mean square deviation of surface topography | $S_q$ | |
| 13. Skewness of topography height distribution | $S_{sk}$ | |
| 14. Kurtosis of topography height distribution | $S_{ku}$ | |
| 15. Maximum peak height of the surface topography | $S_p$ | three-dimensional |
| 16. Maximum valley depth of the surface topography | $S_v$ | |
| 17. Maximum height of surface topography | $S_z$ | |
| 18. Arithmetical mean deviation of surface roughness | $S_a$ | |

*2.3. Surface Free Energy*

For the determination of the SFE, contact angle measurements were performed for each tested material and for each of the four surface roughness levels—the specimen was cleaned with distilled water and isopropanol; then, 1-µL of distilled water and methylene iodide were applied to the specimen's surface. Within 30 s after application, a computer-aided measurement device (Drop Shape Analyzer DSA25, Krüss, Hamburg, Germany) performed ten contact angle measurements for each liquid. The SFE was calculated using the formula introduced by Owens and Wendt [30].

*2.4. Microbial Culture*

The test microorganisms of *C. albicans* (lot no. 1386, Deutsche Sammlung von Mikroorganismen und Zellkulturen (DSMZ), Braunschweig, Germany) and *S. sanguinis* (lot no. 20068, DSMZ) were cultured under standard conditions in Universal Medium for Yeast (lot no. 186, DSMZ) or Tripticase Soy Yeast Extract Medium (lot no. 92, DSMZ). Both microorganisms were harvested by centrifugation, washed twice with phosphate-buffered saline (PBS, Merck, Darmstadt, Germany), and resuspended in PBS. The suspensions of *C. albicans* or *S. sanguinis* in PBS were adjusted to have an optical density of 0.3 at 600 nm by densitometry (Bio Photometer, Eppendorf, Hamburg, Germany).

Microbial Adhesion via Luminescence Assay

Under sterile conditions, the specimens were transferred to 24-well plates and attached to well-bottoms using silicone (Z-Dupe, Henry Schein Dental, Langen, Germany). Then, 1 mL of the *C. albicans* or *S. sanguinis* suspension was added to the wells and was incubated for 2 h at 37 °C and 55 rpm. The viable cells were quantified using an adenosine triphosphate (ATP)-based luminescence assay (LT07-221, Lonza, Cologne, Germany)—after washing with PBS twice to remove non-adherent cells, 300 μL of a cell lysis reagent were added to each well to extract ATP. After 10 min, 100 μL of the supernatants were transferred to a 96-well plate, where 100 μL of ATP monitoring reagent plus were added to each well. After 5 min of incubation, the luminescence was measured using a plate reader (FLUOstar Omega, BMG Labtech, Ortenberg, Germany). Standard glass specimens (Paul Marienfeld, Lauda-Koenigshofen, Germany) served as the controls.

### 2.5. Statistical Analysis

#### 2.5.1. Analyses of SFE

The SFE was analyzed using paired comparisons of SFE means of the roughness levels for each test group. The *p*-values were adjusted using the Bonferroni method.

#### 2.5.2. Analyses of Adhesion

The adhesion was analyzed using a two-factorial repeated measures analysis of variance (ANOVA) with interactions. The fixed effects were the materials (YTZP, PEEK, PMMA, and Ti) and roughness. If a significant interaction was observed between the roughness and material, stratified analyses were performed, using Bonferroni-adjusted *p*-values to control the overall type-I error rate. Contrast-tests were conducted for pairwise comparisons, if the null-hypothesis on a main effect was rejected.

#### 2.5.3. Analyses of Roughness Parameters

The $R_a$ means and standard deviations were calculated for each roughness level. Pair-wise Spearman rank correlation coefficients were calculated among the roughness parameters, and the corresponding *p*-values were Bonferroni-adjusted.

#### 2.5.4. Combined Analyses

To compare the eighteen roughness parameters with adhesion, association was modeled separately for each parameter, and the resulting models were compared. We substituted the level of roughness in the adhesion data-set with the level-specific means of the roughness parameters and applied a one-factorial linear mixed model for each material. The resulting models were compared by the goodness of fit ($R^2$) and the corresponding confidence intervals. In this analysis, only the roughness parameters with means ordered consistently with the levels of roughness were considered.

All of the statistical analyses were performed with R software (version 3.4.0, R Core Team 2018).

## 3. Results

### 3.1. Surface Roughness

After surface modifications, the $R_a$ values of the four test groups were measured. Table 4 shows that the mean $R_a$ values of the four roughness levels were in accordance with the intended $R_a$ values from Table 2. Figure 1 shows the confocal images of the tested materials' surfaces at the four roughness levels.

**Table 4.** Mean $R_a$ values of the four roughness levels ($n = 15$); sd = standard deviation.

| Roughness Level | $R_a$ (µm) Mean (sd) | | | | |
|---|---|---|---|---|---|
| | All Materials | YTZP | PEEK | PMMA | Ti |
| I | 0.1 (0.1) | 0.1 (0.1) | 0.1 (0.1) | 0.1 (0.1) | 0.0 * (0.0) |
| II | 0.2 (0.1) | 0.2 (0.1) | 0.2 (0.1) | 0.2 (0.0) | 0.3 (0.1) |
| III | 0.9 (0.2) | 0.9 (0.1) | 0.9 (0.1) | 0.8 (0.2) | 0.8 (0.3) |
| IV | 1.8 (0.3) | 1.9 (0.2) | 2.0 (0.1) | 1.7 (0.3) | 1.9 (0.2) |

* 0.03.

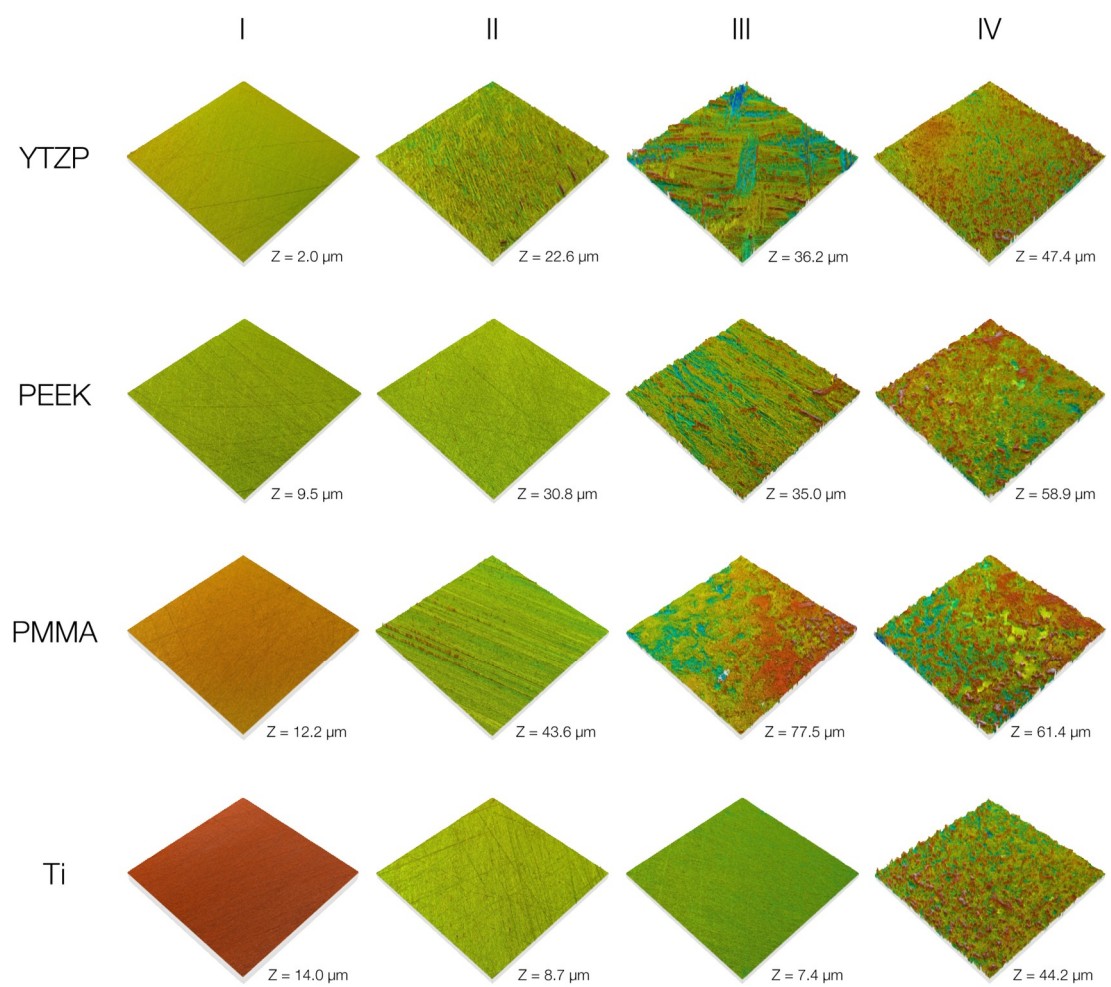

**Figure 1.** Confocal images of the tested materials standardized zirconia (YTZP), polymethylmethacrylate (PMMA), polyetheretherketone (PEEK), and titanium alloy (Ti) at the four roughness levels (I–IV) for 500 µm × 500 µm scan areas; Z = z-axis dimension.

### 3.2. Surface Free Energy

Paired comparisons within each test group revealed that the SFE did not change significantly between the four roughness levels (data not shown).

### 3.3. Microbial Adhesion

The luminescence assays showed that the *C. albicans* adhesion to the materials increased with higher $R_a$ values (Figure 2a). ANOVA indicated significant differences in the *C. albicans* adhesion among the $R_a$ values; paired comparisons showed that the *C. albicans* adhesion to YTZP increased significantly between roughness levels II and IV. Adhesion to PEEK increased significantly between

levels I and IV, while adhesion to PMMA showed a significant increase between roughness levels I and III, II and IV, and III and IV. Adhesion to Ti decreased significantly between roughness levels I and II, but increased significantly between levels I and IV, and between levels II and III.

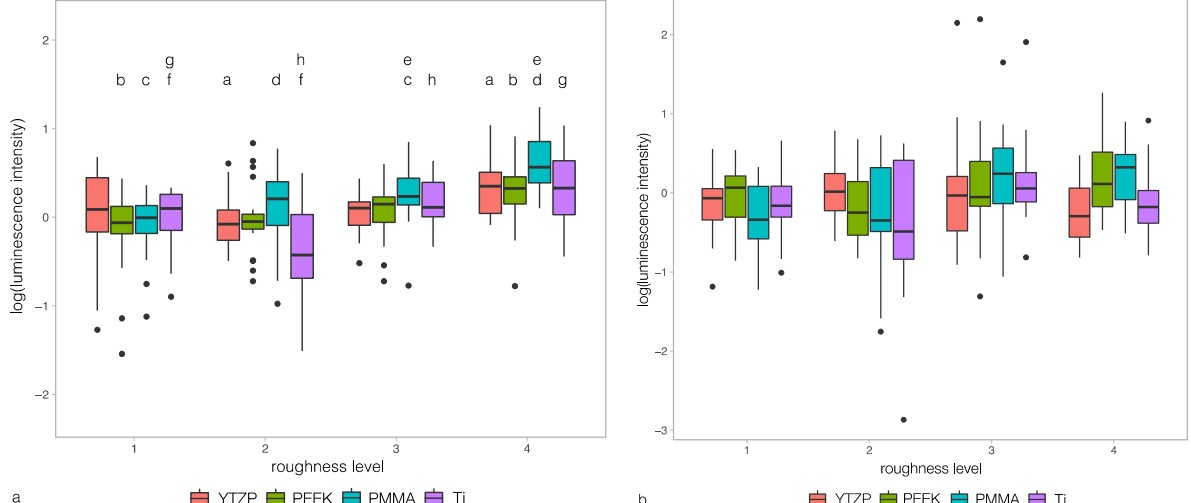

**Figure 2.** Relative adhesion of (**a**) *C. albicans* and (**b**) *S. sanguinis* to the test groups at four different roughness levels after 2 h of incubation. Levels of relative luminescence intensities from the adenosine triphosphate (ATP) assays correlate to the number of viable cells; relative luminescence intensities are shown logarithmized. Bars labeled with identical letters are significantly different at α = 0.05.

Unlike *C. albicans*, *S. sanguinis* adhesion showed no overall trend to increase with the increasing $R_a$ values. A statistical analysis of the adhesion revealed no significant differences among the roughness levels for any material (Figure 2b).

### 3.4. Association of the Roughness Parameters with Microbial Adhesion

We measured 18 roughness parameters for the test groups at four roughness levels. The roughness parameters were fitted with the measured adhesion values using mixed linear models. $S_a$ has the best correlation with *S. sanguinis* adhesion for all of the materials. For the *C. albicans* adhesion, $S_a$ showed the highest goodness of fit only for Ti; for YTZP, PEEK, and PMMA, $R_{dc}$ was highly associated with the *C. albicans* adhesion. Compared to the respective goodness of fit and the confidence intervals of $R_a$, neither $S_a$ nor $R_{dc}$ differed significantly from $R_a$ (Table 5).

**Table 5.** Significance and goodness of fit ($R^2$) of the fitted models for the parameters that showed the highest $R^2$, and for the $R_a$, separated by microorganism and material. LCL = lower confidence level, UCL = upper confidence level, Sig. = significance (no = levels of both parameters are not significantly different).

| Microorganism | Material | Parameter | $R^2$ | LCL | UCL | Sig. |
|---|---|---|---|---|---|---|
| *C. albicans* | YTZP | $R_{dc}$<br>$R_a$ | 87<br>80 | −23<br>−27 | 197<br>187 | no |
| | PEEK | $R_{dc}$<br>$R_a$ | 127<br>123 | −1<br>−3 | 255<br>249 | no |
| | PMMA | $R_{dc}$<br>$R_a$ | 308<br>297 | 150<br>140 | 466<br>454 | no |
| | Ti | $S_a$<br>$R_a$ | 146<br>135 | 12<br>5 | 280<br>265 | no |

**Table 5.** *Cont.*

| Microorganism | Material | Parameter | $R^2$ | LCL | UCL | Sig. |
|---|---|---|---|---|---|---|
| *S. sanguinis* | YTZP | $S_a$ | 2 | −16 | 20 | no |
| | | $R_a$ | 2 | −16 | 20 | |
| | PEEK | $S_a$ | 42 | −39 | 123 | no |
| | | $R_a$ | 39 | −39 | 117 | |
| | PMMA | $S_a$ | 157 | 20 | 294 | no |
| | | $R_a$ | 153 | 17 | 289 | |
| | Ti | $S_a$ | 22 | −38 | 82 | no |
| | | $R_a$ | 19 | −36 | 74 | |

### 3.5. Correlation Analysis of the Roughness Parameters

The roughness parameters determined in the present study were correlated using pair-wise Spearman rank correlation coefficients ($r$). From the vast amount of acquired data, the representative parameters were summarized graphically in Figure 3. $S_a$, $R_{dc}$, and $R_z$ were positively correlated with each other and with $R_a$ ($r$ values near 1); opposite to this, $R_{mr}$ and $S_{sk}$ showed no correlation with $R_a$ or any of the other shown parameters (low $r$-values).

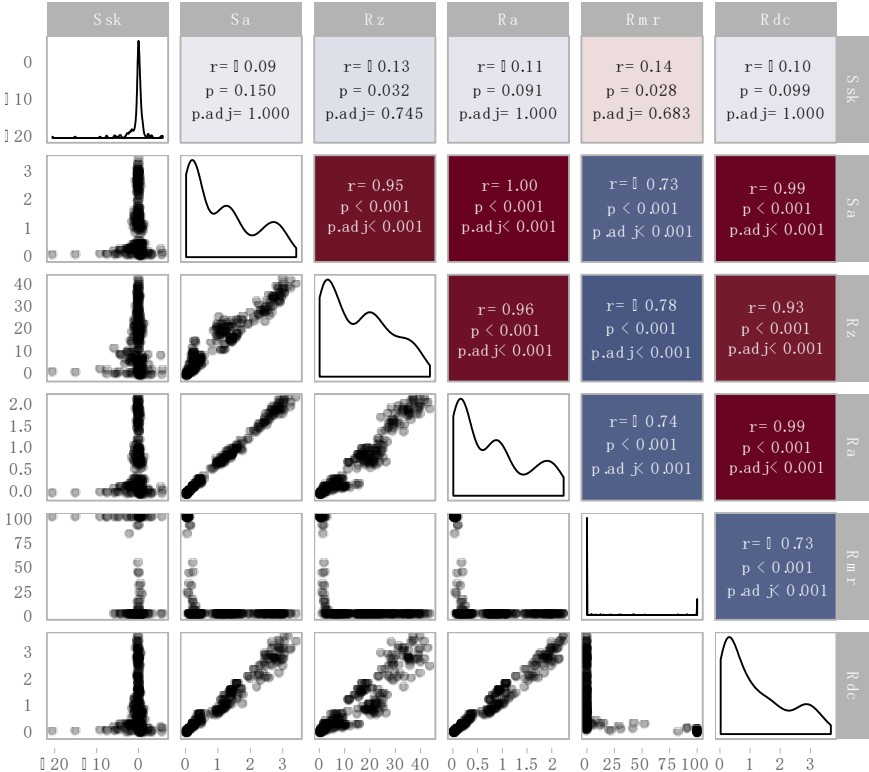

**Figure 3.** Correlation matrix for representative roughness parameters with scatterplots of the roughness values. $S_a$, $R_{dc}$, and $R_z$ are positively correlated with each other and with $R_a$; $R_{mr}$ and $S_{sk}$ show no linear correlation with $R_a$ or any of the other shown parameters. $r$ = Pearson correlation coefficient; $p$ = $p$-value; $p$. adj = Bonferroni-adjusted $p$-value.

## 4. Discussion

The results of the present study suggest partial acceptance of the initial hypothesis. The $R_a$ and 17 other roughness parameters were evaluated for their predictability for initial *C. albicans* and *S. sanguinis* adhesion to four dental materials at four different roughness levels. As hypothesized, a three-dimensional surface parameter, namely $S_a$, showed a higher correlation with *C. albicans*

and *S. sanguinis* adhesion than the generally used two-dimensional profile parameter $R_a$. Yet, the demonstrated differences were not significant because of the overlapping confidence intervals. Moreover, our hypothesis needs to be partially denied, as $R_{dc}$, a two-dimensional profile parameter, was the most appropriate to predict *C. albicans* adhesion to PEEK, PMMA, and YTZP. $R_{dc}$ is a functional parameter that describes the profile section height between two material ratios, and is commonly used in the automotive industry, where it excludes the highest peaks that will be worn out, and the deepest valleys that will be filled during the running-in phase of new engines and bearings [31]. While $S_a$ is suggested to be a useful parameter for biomaterial discrimination at the microscale [20], to our knowledge, there is no literature available on $R_{dc}$ and its suitability in this regard. Because of the insignificant differences to $R_a$, we consider the high association of $R_{dc}$ with *C. albicans* adhesion to be coincidental in our experimental setting.

Generally, a high positive correlation between the roughness parameters indicates redundancy and low discriminatory powers to characterize surfaces of biomaterials [20]. In the present study, all of the parameters that describe the heights or valleys showed a high positive correlation, except for the roughness parameters that describe distinctive qualities such as the kurtosis ($R_{ku}$ and $S_{ku}$), skewness ($R_{sk}$ and $S_{sk}$), or material component ($R_{mr}$) of a profile or a surface. The results are in line with previous studies that state that many of the parameters used to characterize surfaces of biomaterials are redundant [21,32].

Interestingly, the profile and surface parameters correlated. This indicates that the surface treatment with an automated grinding machine allowed for consistent surfaces with a high comparability in terms of profile and surface quality.

The high positive correlations between most of the roughness parameters and their overlapping confidence intervals explain why none of the parameters were significantly more suitable to predict the initial microbial adhesion than $R_a$. The low discriminatory powers of the surface roughness parameters in the present study are opposed to a study that stated that $S_a$ and $S_z$ are appropriate parameters to discriminate between the polished surfaces of dental materials [20]. If the profile and surface roughness parameters of a material show more discriminatory powers than in the present study, parameters other than $R_a$ will be associated significantly more with initial microbial adhesion. In terms of clinical relevance, future in vitro research could compare the predictability for initial microbial adhesion to prefabricated materials, such as implants with sandblasted or coated surfaces, by profile and surface roughness parameters.

The materials examined in this study were chosen because of their relevance in prosthetic rehabilitation, and their exposure to the oral microbiome in vivo. Well-established dental materials like titanium alloy and PMMA were examined, which are considered the gold standard for implants and implant abutments or denture bases, respectively [33–36]; yttria-stabilized zirconia is increasingly meaningful in implant and abutment technology, and PEEK shows broad indications for fixed and removable prostheses [37,38]. Clinically, both materials are biocompatible and have metal-free appearances, but they need further in vitro and in vivo examination [37–40]. These considerations justify the material selection in the present study.

To evaluate the predictability for the initial *C. albicans* and *S. sanguinis* adhesion by various roughness parameters, the materials were tested with four roughnesses. An $R_a$ value of 0.2 μm is suggested as the threshold below which roughness does not further affect bacterial adhesion to the titanium alloy [23,41]; above this value, the roughness influences the bacterial adhesion [42]. Considering this, in the present study, one surface treatment of the tested materials was set below the threshold ($R_a$ <0.1 μm), one at the threshold ($R_a$ ~0.2 μm), and two surface treatments were chosen considerably above the threshold ($R_a$ = 0.7–1 μm and $R_a$ = 1.7–2 μm). *C. albicans* and *S. sanguinis* showed differing adhesion properties for the materials and the variety of surface roughnesses. Overall, the *C. albicans* adhesion increased with the increasing $R_a$ values, which is in line with other in vitro studies [43,44]. Confirmatory for the threshold, no significant differences were detectable between $R_a$ <0.1 μm and $R_a$ ~0.2 μm for *C. albicans* adhesion to zirconia, PEEK, and PMMA. Yet, *C. albicans* adhesion

to the titanium alloy was significantly higher for $R_a$ <0.1 μm than for $R_a$ ~0.2 μm. Our findings indicate that the $R_a$ threshold, originally suggested for bacterial adhesion to titanium alloy abutments [40], is material-dependent for yeast adhesion, and needs reevaluation.

Their fundamental morphological and biochemical differences equip fungi and bacteria with differing capabilities in order to adhere to identical substrates [45]. *S. sanguinis* adhesion showed no tendency to increase with the increasing $R_a$ values. This finding is in line with a study showing that *S. sanguinis* adhesion to enamel did not increase with increasing roughness [46]. Contrariwise, *S. sanguinis* adhesion to dentine root surfaces is positively correlated with roughness [47]. Possibly, *S. sanguinis*, being a small-sized early colonizer in oral biofilms [6], is less demanding and more robust in terms of surface modalities than the larger yeast *C. albicans* in the present study. The conflicting evidence about *S. sanguinis* adhesion and its correlation with surface roughness needs further clarification.

Although the SFE is frequently suggested as less significant for the microbial adhesion to a material than the roughness [17,42], for some materials, the SFE may be even more influential than the roughness [11,48]. As roughening procedures may alter a material's SFE, and, consequently, the microbial adhesion [49], the SFE was considered as a possible confounder affecting the *C. albicans* and *S. sanguinis* adhesion in the present study. Contact angle analysis showed that rougher surfaces did not significantly change the SFE. Therefore, the SFE had a negligible influence on *C. albicans* and *S. sanguinis* adhesion, and did not constitute a study bias.

Within the limitations of an in vitro study, it was not beneficial to use roughness parameters other than $R_a$ to predict the initial *C. albicans* and *S. sanguinis* adhesion to dental materials. However, some parameters showed a higher, but statistically insignificant, correlation with microbial adhesion than $R_a$. $R_a$ is well-established, easy to determine, and reproducible to predict the initial microbial adhesion to dental biomaterials with polished surfaces.

We conclude that *C. albicans* and *S. sanguinis* have different adhesion properties for different surface roughnesses, especially below the $R_a$ of 0.2 μm. The $R_a$ threshold, below which roughness does not further influence adhesion, is different for yeast and bacterial adhesion, and needs further in vitro and in vivo elucidation.

**Author Contributions:** Conceptualization, R.B. and T.W.; Methodology, T.W. and M.H.; Validation, A.S. and T.W.; Formal Analysis, A.S. and S.K.; Investigation, M.H. and O.K.; Resources, R.B.; Data Curation, A.S. and M.H.; Writing (Original Draft Preparation), A.S.; Writing (Review and Editing), R.B., T.W. and S.K.; Visualization, A.S. and T.W.; Project Administration, T.W.; Supervision, R.B.

**Funding:** We acknowledge support by the German Research Foundation and the Open Access Publication Funds of the Goettingen University.

**Acknowledgments:** We would like to thank Christoph Anten for his contribution to the statistical analyses.

**Conflicts of Interest:** The authors declare no conflict of interest.

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
