# Peer review of "Predictability of Microbial Adhesion to Dental Materials by Roughness Parameters"

_coatings, doi:10.3390/coatings9070456_

Round 1
Reviewer 1 Report
This is a very interesting and well prepared manuscript. Excellent work, well done. I do not have any suggestions.
Author Response
Thank you for your kind review!
Reviewer 2 Report
This paper on adhesion was interesting and well constructed. The writing could be more concise and direct to increase its impact. Please see comments.

Author Response
(please also see the attachment)
Thank you for your comprehensive review, which contains many helpful improvements to our manuscript. We included your language corrections and are pleased to see that our manuscript is less wordy and more concise now. The comments you made are addressed point by point in order of appearance in the following:
1. To avoid confusion with the element titanium, we introduced the abbreviation ‘Ti’ for titanium alloy in table 1 as well as in the abstract and in the introduction. The manuscript now reads: ‘After four surface roughness modifications using standardized polishing protocols, zirconia, polymethylmethacrylate, polyetheretherketone and titanium alloy specimens were characterized by the Ra and 17 other parameters using confocal microscopy. (...) Surfaces of standardized zirconia (YTZP), polymethylmethacrylate (PMMA), polyetheretherketone (PEEK) and titanium alloy(Ti) specimens were characterized according to those parameters using confocal microscopy after four different surface roughness modifications.’ Wherever ‘titanium’ was used in the text, we added ‘alloy’ for clarity. We hope these alterations to the manuscript meet your request adequately.
2. As requested by you, we defined the Sa and the Rdc in the abstract: ‘Although the Sa, i.e. the arithmetical mean deviation of surface roughness, and the Rdc, i.e. the profile section height between two material ratios, showed higher correlations with the microbial adhesion than the Ra, these differences were not significant.’
3. We removed the word ‚most’ from the introduction. The respective passage now reads as follows: ‚Oral biofilms are responsible for mostdental pathologies such as caries, periodontitis, peri-implantitis, denture stomatitis or candidiasis [1-4].’
4. As recommended by you, we included the statement from the discussion in the introduction: ´In the literature, many parameters describe surface roughness, some of which are suitable to discriminate between polished surfaces of dental materials[18-20].´
5. In the introduction, we rephrased the passage about the parameter rash. It now reads as follows: ‘In the literature, many parameters describe surface roughness [18-20] and their rapidly increasing number was called “the parameter rash” by Whitehouse in 1982 already [21].’
6. We stumbled across a multitude of surface parameters other than the Ra that are used outside of biomedical research. We wondered if these parameters are more appropriate to predict microbial adhesion. To our knowledge, there is no literature proving any parameter’s superiority.
7. We adjusted Table 1 according to your annotations.
8. As suggested, in the methods section we clarified that the specimens were tested successively by rephrasing the passage as follows: ‚The same specimens were tested successively: after completing adhesion testing for one roughness level, specimens were rinsed with distilled water and 70 % ethanol, dried and polished to the next lower Ra value level.‘
9. In the methods section, we rephrased the passage about the ISO classification using your helpful recommendation. It now reads: ‘These parameters were chosen based on ISO 4287 for two-dimensional and ISO 25178 for three-dimensional parameters.’
10. The roughness classification from Table 2 does not relate to an ISO classification. We explain the chosen values in the discussion section: ‘An Ra value of 0.2 µm is suggested as the threshold below which roughness does not further affect bacterial adhesion to titanium alloy [23,40]; above this value, roughness influences bacterial adhesion [41]. Considering this, in the present study, one surface treatment of the tested materials was set below the threshold (Ra < 0.1 µm), one at the threshold (Ra ~ 0.2 µm) and two surface treatments were chosen considerably above the threshold (Ra = 0.7 – 1 µm, Ra = 1.7 – 2 µm).’
11. Moreover, we centered the columns as recommended by you.
12. As requested, we numbered the parameters in Table 3.
13. In the methods section, we added the cleaning procedure we used before the SFE was determined: ‘(…) the specimen was cleaned with distilled water and isopropanol; then,one µl of distilled water and methylene iodide were applied to the specimen’s surface.’
The new paragraph in the methods section (‘Analyses of roughness parameters’) was removed.
14. Thank you for this annotation, we used a superscript number: ‘The resulting models were compared by the goodness of fit (r2) and the corresponding confidence intervals.’
15. Yes. Of the 18 parameters that we initially measured, only those that showed a positive correlation with the four roughness levels were used for further adhesion correlation analyses. We hope this explanation answers your question adequately.
16. In Table 4, we centered the columns. Also, we exchanged the log(Ra) values for Ra values.
In the results section, we removed the redundant passage about the surface free energy. Thank you for this helpful remark.
17. We did not show any contact angle data in the manuscript in order to keep the amount of included tables and figures as compact and concise as possible. As mentioned in the results section, the acquired data did not show significant differences between the roughness levels. For clarification, we adjusted the passage indicating that the data is not shown: ‘Paired comparisons within each test group revealed that the SFE did not change significantly between the four roughnesslevels (data not shown).’
18. Thank you for this remark. We moved the sentence up within the paragraph and rephrased it as follows: ‘The Rdc is a functional parameter that describes the profile section height between two material ratios and is commonly used in the automotive industry, where it excludes the highest peaks that will be worn out and the deepest valleys that will be filled induring the running-in phase of new materialsengines and bearings[29].’
Please see above.

Reviewer 3 Report
Review of the manuscript entitled: “Predictability of microbial adhesion to dental Materials by roughness parameters”.
In this study, Authors investigated predictability of two and three-dimensional roughness parameters for initial adhesion of C. albicans and S. sanguis.
The manuscript is well organized and clearly written. Some details must be considered to achieve the publication in this Journal.
1. From results, it emerges that no parameter describing the surface roughness is equally correlated to adhesion of both C. albicans and S. sanguis. These microrganisms are very different from each other. In the discussion, aspects related to the capability of microorganism to differently adhere to the substrates should be treated more deeply.
2. Authors investigated materials (zirconia, polymethylmethacrylate, titanium) that have very different clinical applications (implants, prosthesis..) and that may be involved in various diseases. It should be investigated if for each material there is a roughness parameters that is more suitable to predict the microbial adhesion. Furthermore, in the study, the choice of the investigated microbes should be more calibrated on the clinical purpose of the tested material (i.e. studing specific bacteria involved in peri-implant disease on titanium...).
Author Response
Thank you for your kind review.
As requested, we included the capability of microorganisms to differently adhere to the substrates in the discussion section: ‘Their fundamental morphological and biochemical differences equip fungi and bacteria with differing capabilities to adhere to identical substrates.’
1. We did investigate the most suitable roughness parameter in terms of prediction for S. sanguinis and C. albicans adhesion for each tested material separately. The results of this investigation are summarized in table 5. Does this answer your request adequately?
2. We agree with your remark that it would be interesting to investigate disease-specific microbes for peri-implantitis etc.. However, in the present study, we chose S. sanguinis as an exemplary and representative bacterium that is an early colonizer and thus relevant for initial biofilm formation to all of the tested materials. C. albicans was chosen because we, as prosthodontists, are interested in fungal adhesion to denture bases due to the clinical relevance in stomatitis that we are confronted with very often. In a next step, we are planning to investigate microbial adhesion more disease-specific, as recommended by you. We hope this answer addresses your concerns satisfyingly.
Reviewer 4 Report
The paper presents an interesting analysis of microbial adhesion on several substrates with different surface finishing. Adhesion was evaluated throughout luminescence intensity while roughness was assessed by 17 parameters.
1. The first sentence of the abstract is a very strong one: “Microbial adhesion to intraoral biomaterials is correlated with surface roughness”. This idea is repeated in the introduction as …”microbial adhesion correlates to its surface roughness (topography) and its surface free energy (SFE) (hydrophobicity) after manufacturing processes [3]”.
The word Correlation implies that a statistical analysis was performed.
Taking a look at reference [3], the above statement is not so direct as authors mention “an increase in surface roughness above the Ra threshold of 0.2 mm and/or of the surface-free energy facilitates biofilm formation on restorative materials.” From a brief glance I did not find in reference [3], a statically analysis that allows to mention that there is a correlation. Also the reference [3] is a review paper.
Please clarify and provide a reference with a statistical treatment where the correlation between adhesion and surface roughness is shown.
2. In the abstract there is no definition of Sa and Rdc
3. Instead of “to predict microbial adhesion to dental materials” use “to predict the tendency for microbial adhesion to dental materials. “
4. Was R the software used for the Bonferroni method as well as for the rest?
5. Table 4 –Mean values of how many measurements of Ra?
6. Adhesion is a property that may be quantified by several techniques and parameters. Authors use properties and parameters indistinctly, which is not correct. Figure 2 shows log(adhesion) versus roughness level. Why use of the word adhesion and not use LI which denotes luminescence intensity? Authors need to clarify what is the meaning of a, b, c, …g in figures.
7. Caption of Figure 2: there is a missing word/letter, probably p=0.05 ?
8. Conclusions: Ra is a sufficient roughness parameter in terms of prediction for initial microbial adhesion. This sentence is wise, resumes the findings of the paper and is not so strong as the one of comment 1.
Author Response
Thank you for your helpful and well-structured review of our manuscript. We answered to your annotations point by point in the following:
1. We attenuated the first sentence of the abstract by changing it to the following: ‚Microbial adhesion to intraoral biomaterials is correlatedassociated with surface roughness.
2. We included definitions of Sa and Rdc in the abstract: ‚Although the Sa, i.e. the arithmetical mean deviation of surface roughness,and the Rdc, i.e. the profile section height between two material ratios,showed higher correlations with the microbial adhesion than the Ra, these differences were not significant.’
3. Thank you for this helpful remark. We changed the sentence accordingly: ‚The results of the present study will outline which surface roughness parameters are most appropriate in order to predict the tendency for microbial adhesion to dental materials.
4. Yes, R software was used for all statistical analyses.
5. In the Materials and Methods section, we mention ‚Ra values were calculated for five specimens at three sites of each tested material via confocal microscopy (Zeiss Smartproof 5, Carl Zeiss Microscopy, Jena, Germany) and automated software analysis (Confomap, Carl Zeiss Microscopy).’ To clarifiy this for the reader, we included ‚n = 15’ in the heading of table 4: ‚Mean Ra values of the four roughness levels (n = 15); sd = standard deviation.
6. As recommended by you, we renamed the y-axis in figure 2 to ‘log(luminescence intensity)’. Thank you for this helpful annotation. We clarify the meaning of the letters in figure 2 by stating ‘Bars labeled with identical letters are significantly different at a= 0.05.’ We chose this way to indicate significances because asterisks would have made the figure layout very confusing, in our opinion. We hope this explanation addresses your remark adequately.
7. The caption now reads as follows: ‚Bars labeled with identical letters are significantly different at a= 0.05.’ Thank you for this remark.
8. We are glad to hear that you agree with the conclusions we drew.
Round 2
Reviewer 3 Report
Authors solved all concerns and the manuscript is now publishable.